# The Effect of Cardiorespiratory Exercise in the Prevention and Treatment of Hypertension among HIV-Infected Individuals on Antiretroviral Therapy in Mthatha, South Africa

**DOI:** 10.3390/healthcare11131836

**Published:** 2023-06-23

**Authors:** Urgent Tsuro, Kelechi Elizabeth Oladimeji, Guillermo-Alfredo Pulido-Estrada, Teke Ruffin Apalata

**Affiliations:** 1Department of Public Health, Faculty of Health Sciences, Walter Sisulu University, Mthatha 5100, Eastern Cape, South Africa; tsurourgent@gmail.com (U.T.); oladimejikelechi@yahoo.com (K.E.O.); gpulidoestrada@wsu.ac.za (G.-A.P.-E.); 2College of Graduate Studies, University of South Africa, Johannesburg 0001, Gauteng, South Africa; 3Department of Laboratory Medicine and Pathology, Faculty of Health Sciences, Walter Sisulu University, Mthatha 5100, Eastern Cape, South Africa

**Keywords:** cardiorespiratory fitness, concurrent training, hypertension, HIV, black people, randomized control trial

## Abstract

The prevalence of hypertension among people living with HIV (PLHIV) on antiretroviral therapy (ART) is concerning. Physical activity is a proposed approach for managing and avoiding hypertension in this population. While cardiorespiratory exercises (CET) have been efficacious in the general population, its effectiveness in PLHIV on ART, especially in the study setting, in Mthatha is unknown. Consequently, the purpose of this study was to see if CET improves cardiorespiratory fitness in HAART-treated PLHIV with blood flow restriction (BFR) in Mthatha, South Africa. A quasi-experimental study with 98 participants (49 of whom were cases) was carried out. Cases were participants assigned to the CET intervention group that comprised of concurrent training as it included both aerobic and resistance exercise, matched on age and gender. The relationship between CET and hypertension was assessed using logistic regression after adjusting for possible confounding variables. At baseline, there was no significant difference between the physical characteristics of the two groups, and after the intervention, there was a significant difference. Obesity and central adiposity were identified as strong risk factors for hypertension. The findings also indicated that a reduction in waist circumference and body mass index had a significant positive association with hypertension treatment amongst the intervention group (*p* < 0.05). According to the results of the study, CET has the potential to be an efficient and economical non-pharmacological intervention for the management and control of hypertension in PLHIV. However, further study is required to establish how long, how intense, and what kind of exercise is best for this population.

## 1. Introduction

Hypertension is a common health problem that affects a significant proportion of the global population and is a leading risk factor for cardiovascular disease (CVD) including stroke, and other non-communicable diseases (NCDs) [1]. Human immunodeficiency virus (HIV) infection and antiretroviral therapy (ART) use have been shown to contribute to hypertension by causing endothelial dysfunction, arterial stiffness, and inflammation [2]. People living with HIV (PLHIV) who are on ART are at a higher risk of developing hypertension than the general population [3,4,5]. As an indisputable preventative risk factor for fatalities both globally and in relation to CVD, hypertension has emerged as a foremost cause of concern [6,7]. According to recent statistics, the year 2010 was marked by a staggering 31.1%, or approximately 1.39 billion individuals worldwide, who possessed hypertension, also characterized as systolic BP values greater than or equal to 140 mmHg coupled with diastolic BP values of no less than 90 mmHg [8]. While it can be attributed to diverse factors such as aging demographics or decreased level of physical activity across unique cultures, hypertension prevalence continues its upward trajectory on a global basis [8].

Being overweight or obese has become increasingly common among people living with HIV (PLHIV) over the past two decades, with rates that frequently outstrip regional population trends and, in some cases, national averages [9,10]. Studies show that this rise in cases can be traced back to both the introduction of new ART agents and advancements in survival rates [10]. Obesity can lead to grave health concerns that have the potential of being life-threatening. Among these, hypertension [11,12] is a serious issue. In their study, Cheah et al. [13] suggested an affirmative correlation between being overweight or obese and the incidence of hypertension. The researchers found that people with unhealthy waist circumferences and waist-to-hip ratios, a high conicity index, and excess fat in their bodies were at greatest risk. The findings of Mehdad et al. [14] lend further credence to this contention, showing that a higher body mass index (BMI) is associated with an increased risk of hypertension. High levels of abdominal fat are associated with an increased risk of cardiometabolic disease [15]. Disturbingly, research indicates a positive correlation between visceral fat and the remarkable body shape index (ABSI) [16], while the body adiposity index (BAI) seems more adept at mirroring weight reduction in individuals experiencing class I and II obesity [17]. 

Skinfold thickness (ST) is an easily obtained obesity index and seems like a good way to estimate the subcutaneous fat [18]. Both subscapular skinfold thickness (SST) and triceps skinfold thickness (TST) have been shown to be positively associated with elevated blood pressure (BP) and hypertension in various populations, especially adolescents, while relevant studies from adults are limited [19,20]. 

Over the course of the last 20 years, nations with prosperous economies have seen a small decline in people suffering from high blood pressure, whereas low- and middle-income countries (LMICs) have witnessed noteworthy rises [21]. The World Health Organization (WHO) reports that the continent of Africa has the highest prevalence of hypertension worldwide. A staggering statistic reveals that more than 46% of adult Africans struggle with this disease [22,23].

Hypertension is brought about by several factors that can be labeled as either modifiable or non-modifiable risk factors [24]. Among the non-modifiable variables are gender, age, race, and genetic predisposition which cannot be altered with much control [25]. Meanwhile, risk factors such as diet patterns, physical inactivity, smoking habits, obesity levels, and high stress situations, to name a few examples, are considered modifiable due to their dependence on individual lifestyle choices. Additionally, certain medications and excessive consumption of alcohol have also been correlated with hypertension [26].

In most developing countries, cases of body fat redistribution (BFR) and metabolic imbalances have been documented in HIV patients who follow a highly active antiretroviral therapy (HAART) treatment plan [27]. Similarly, these occurrences are prevalent in African individuals with HIV who receive their first-line WHO-recommended HAART regimen [28]. With the increased accessibility of HAART therapy for those living with HIV in resource-limited regions worldwide [29,30], as well as improvements to their quality of life [31], managing hypertension has become a challenge. 

However, physical exercise is a powerful therapy that does not involve the use of drugs and can help prevent and manage high blood pressure [32]. Among the various types of physical activities, cardiorespiratory exercise training (CET) deserves special mention as it has demonstrated its efficacy in lowering blood pressure levels in hypertensive individuals while simultaneously increasing cardiovascular fitness and, by extension, reducing the risk of developing hypertension [33]. CET holds great relevance for PLHIV because, beyond its proven effectiveness in treating hypertension, it also facilitates improvements in lipid imbalances [31], enhances insulin sensitivity [30], and helps to eliminate abdominal adiposity or trunk fat gain; all these factors combine toward creating a considerably improved cardiovascular risk profile. This time-honored lifestyle habit is known for its cost-effectiveness relative to pharmacological interventions while achieving an equally significant impact [34].

Several non-randomized controlled trials of aerobic and resistance exercise studies with small sample sizes and short training durations have reported improved body composition profiles in PLHIV with BFR in Western countries [35]. In resource-limited areas such as sub-Saharan Africa, CET may be a particularly important treatment for BFR in PLHIV taking HAART. In one study conducted to examine the effect of aerobic exercise interventions on cardiopulmonary and psychological parameters in PLHIV, it was found that aerobic exercise interventions can improve cardiopulmonary fitness and psychological status in this population [36]. Another study by Barros, et al. [37] that sought to investigate the effects of aerobic exercise on blood pressure in HIV-infected individuals, aerobic exercise significantly reduced blood pressure in HIV-infected individuals. We are unaware of any studies that specifically sought to examine the impact of exercise on hypertensive individuals, especially PLHIV within parts of rural Eastern Cape, South Africa, although a small number of studies have investigated this question for hypertensive individuals in other rural areas of South Africa. To determine if CET enhances cardiorespiratory fitness in HAART-treated PLHIV with BFR in Mthatha, South Africa, we conducted a 12-week quasi-experimental study.

## 2. Materials and Methods

### 2.1. Study Design and Setting

With a quasi-experimental study design, this study was conducted between January and April of 2020, and it only included PLHIV with elevated BP who identified as being of African descent. Participants that were enrolled were those who received health care services from health facilities around Mthatha, in the Eastern Cape Province of South Africa. The Eastern Cape Province has a population of about 7 million, making it the third most populous province in South Africa [38]. The HIV burden in Eastern Cape is reported to be 25.2% [39] compared to the national HIV prevalence of 14% [40].

### 2.2. Enrolment Criteria

Eligible participants were enrolled if they were African, hypertensive, HIV positive, on ART, and on anti-hypertensive treatment for more than 15 years. The participants were supposed to have elevated BP, BFR, and be of any gender. Pregnant individuals and patients that were reluctant to participate were excluded from the study. In addition, individuals who had no BFR, were HIV negative, and not black were not included in the study.

### 2.3. Ethical Approval 

The current study followed the Declaration of Helsinki on ethics [41] and it was approved by the ethics committee at Walter Sisulu University (protocol number: 048/2019) and the Eastern Cape department of health (approval number: EC_201907_020). Participants were given a bilingual (English and local language) information sheet prior to completing the written informed consent form. The information sheet detailed the research procedure and participant responsibilities and rights, and provided a point of contact.

### 2.4. Sample Size 

The G power version 3.1 software [42] was used to determine the necessary sample size, where the effect size was 0.55, α = 0.5 and the power was 0.8. The study included a total of 82 participants, with 41 assigned to the intervention group and 41 to the control group. Twenty percent more people were added to account for dropouts, bringing the total number of participants up to 98.

### 2.5. Sampling Procedure 

This research employed a non-probabilistic sampling strategy known as purposive sampling.

### 2.6. Intervention

The exercise intervention consisted of a 12-week CET program, which was supervised by a trained exercise physiotherapist. The program included 3 sessions per week, with each session lasting 45 min. The exercise program was concurrent training as it included both aerobic exercise (such as brisk walking and cycling) and resistance training (such as lifting weights and using resistance bands). Eligible study participants who were allocated to the intervention arm were requested to come to the community hall within the study setting for the CET program. The intensity of the exercise was gradually increased over the 12-week period to ensure that participants were challenged but not overexerted. The number of repetitions per exercise was increased with time. In the first week, there were 10 repetitions. This was consistently increased by 5 repetitions every week until the 12th week of the CET program. Similarly, the amount of weightlifting using 5 kg dumbbells was increased by the physiotherapist as the participants got used to the weight, thus increasing resistance. 

Those in the control group received no intervention but were instructed to continue their usual daily activities throughout the study period.

### 2.7. Data Collection

Research Electronic Data Capture (RedCap), a web-based online survey tool, was used to digitize the questionnaire and consent forms [43]. In-person interviews with trained research staff were conducted using the digitized tool to collect informed consent and data. The digitalized tool was then utilized for the face-to-face interviews conducted by trained research staff to obtain informed consent and data. Baseline demographic data including age, gender, weight, height, and duration of ART were collected from all participants. Blood pressure was measured at baseline, 6 weeks, and 12 weeks using a calibrated automatic blood pressure monitor. Before checking their BP, participants were asked to rest for five minutes and then the readings were taken three times and the mean was then calculated. Skin fold measurements which include triceps, bicep, suprailiac, and subscapular were measured using a caliper [44]. Physical characteristics including anthropometric indices were measured and/or calculated at baseline and after 12 weeks using the below-mentioned formulae. Skin fold or anthropometric indices are used to show the level of obesity in an individual [45].

### 2.8. Definition of Variables 

#### 2.8.1. Hypertension Measure

Professionals in the medical field used electronic monitors to take standardized readings of patients’ blood pressure. Each participant was given five minutes to relax before their blood pressure was taken. The average of three readings taken at five-minute intervals was calculated. People with high blood pressure were identified using the criteria laid out in the joint national committee (JNC-8) Report on the Prevention, Detection, Evaluation, and Treatment of High Blood Pressure as more than 120 mmHg or a diastolic blood pressure (DBP) reading of less than 90 mmHg on a consistent basis [46]. 

#### 2.8.2. Overweight and Obesity Measure

The participants’ weight was measured using a standard beam balance accurate to within 0.1 kg, and their height was measured using a stadiometer calibrated to within 0.1 cm. After calculating body mass index, subjects were categorized in accordance with WHO recommendations [47]. There were two groups defined by BMI cutoffs: those with a BMI of less than 25 and those with a BMI of more than 25. A non-stretchable plastic tape measure was used to measure the waist circumference (WC) and hip circumference (HC) in centimeters, to the nearest 0.1 cm. Skin fold measurements which include tricep, bicep, suprailiac, and subscapular were measured using a caliper [44].

The ABSI Was Calculated Using the Formula Stated Below [48]
$$ABSI=\frac{Waist\;Circumference}{BMI^{2/3}Height^{1/2}}$$
where:BMI = Body mass index (kg/m^2^)

The BAI Was Calculated Using the Formula Stated Below [49]
$$BAI=\frac{Hip\;Circumference\left(\mathrm{cm}\right)}{Height\;\left(m\right)}-18$$

#### 2.8.3. Social, Economic, and Demographic Measure

A WHO stepwise questionnaire was uploaded to RedCap and used in face-to-face interviews to collect data on demographic and environmental factors. Whether or not they smoked cigarettes or drank alcohol, their highest level of education, and their age in years were also collected.

### 2.9. Data Analysis 

The R studio, version 4.2.1 was utilized for in the data analysis. To ascertain how the variables were spread out, we ran a skewness and kurtosis test for normality. Normally distributed variables were summarized using mean standard deviation (SD). The Wilcoxon rank sum test was utilized to compare the two groups’ means on all continuous variables and the Pearson’s Chi-squared test was used to compare categorical variables. The crude odds ratio (OR), 95% confidence interval (CI), and significance levels were calculated using logistic regression analysis to identify independent variables associated with hypertension. Multi-collinearity was identified using a heat map, which included all predictors of outcomes with a *p* value of less than 0.20 from the bivariate analysis. The multivariate logistic regression did not include the variables with a correlation coefficient of >0.7. One-tailed *p*-values 0.05 were considered significant, and the crude OR and adjusted odds ratio (AOR) with a 95% CI were calculated.

### 2.10. Validity and Reliability

The survey was administered in the language most comfortable to the respondents. The research assistants were given a day of training before they went out into the field. The instrument was researcher- and self-administered, with participants completing it in a quiet room set aside by the clinic’s staff. In cases where participants were asked to administer their own questionnaires, the primary researcher made sure they were all filled out each day.

## 3. Results

### 3.1. Sociodemographic Characteristics of Study Participants

In total, 98 PLHIV who had escalated BP participated in the study. In the current study, participants were matched one-to-one based on their age category and gender; thus, both groups had equal numbers of participants. Generally, there were more females (61%) than males that were enrolled into the study. A greater portion of the participants were single (46%) while those who were cohabiting were the least represented group (2%). Most participants had matric (69%) as their highest educational qualification. The majority of the participants (65%) were unemployed. Most participants reported they were not smoking tobacco (88%) and most (80%) of the participants were not consuming alcohol during the time of the study (Table 1).

### 3.2. Physical Characteristics of the Study Participants before and after the Exercise Intervention

Table 2 represents the physical characteristics of the study participants before and after the exercise intervention. There was not much difference among all the variables that were included in the anthropometric data. According to Table 2, the average tricep between the participants assigned to the intervention was 13 mm and to the control group was 14 mm. The average bicep of the control group was slightly higher (7.90 mm) than the intervention (7.67 mm). The control group had a greater average suprailiac (20 mm) compared to the intervention group (19 mm).The intervention group reported a greater BMI (40 kg/m^2^) than the control group (39 kg/m^2^). The intervention group reported a larger waist circumference (102 cm) than the control group (101 cm). The intervention group also reported a greater BAI (34) than the control group (33) and a greater ABSI (0.74) than the control group (0.70). Generally, most variables were similar in both groups. After the exercise intervention, all the variables of the intervention group decreased but there was no significant change among BAI and ABSI.

#### 3.2.1. Comparison of Blood Pressure among the Control and Intervention Groups before and after Exercise

Participants’ blood pressure was also measured before and after the intervention (Figure 1) to see if it changed. Figure 1A shows that, after three months of the study, the median SBP of the control group increased slightly, while the SBP of participants assigned to the intervention decreased significantly, resulting in a median SBP below 120 (the cutoff). The same pattern applied to the DBP; the median DBP of the control group increased after the study period, whereas the DBP of the participants that were assigned to the intervention decreased significantly and the median DBP was below 90, which was the cutoff DBP (Figure 1B).

#### 3.2.2. Comparison of Blood Pressure among the Control and Intervention Groups before and after Exercise, Categorized by the Participants’ Smoking and Drinking Status

We further stratified the BPs of the two study groups by smoking and drinking status. Generally, there was no significant difference in the BP before and after the study among all the groups (Figure 2). The median SBP of all the participants assigned to the intervention group decrease below 120, except for those who drank alcohol and smoked tobacco (Figure 2A). The median DBP of all the participants assigned to the intervention group decrease below 90, except for those who drank alcohol and smoked tobacco (Figure 2B).

### 3.3. Drug-to-Drug Interaction among the Intervention Group

Figure 3 below shows four graphs with the number of participants who were taking hypertension drugs grouped according to the type of ART they were taking. The majority of the study participants did not have hypertension at the end of the study, except for one participant who took 1S3E and amlodipine (Figure 3).

### 3.4. Variable Selection

We constructed a heat map in order to identify multi-collinearity among anthropometric variables. Positively correlated variables are represented by the red color, whereas those that had a negative correlation are in purple. The intensity of the color represents the strength of the association between variables. Multi-collinearity exists when the magnitude of the correlation is greater than 0.7. All variables with correlation greater than 0.7 did not qualify for inclusion into the logistic regression (Figure 4).

### 3.5. Anthropometric Factors Associated with Hypertension among PLHIV

A logistic regression analysis was performed to determine the factors associated with the development of hypertension among the study participants. The reduction in BP was associated with BMI (OR = 1.71; 95% CI: 1.20–2.63; *p* = 0.006) and WC (OR = 1.33; 95% CI: 1.17–1.61; *p* < 0.001) (Table 3).

## 4. Discussion

Even though ART has made a positive impact on people living with HIV/AIDS, it has also contributed to an increase in the prevalence of obesity [10], which is associated with hypertension [11,12]. It is well-known that hypertension poses a significant health risk to PLHIV, especially those receiving ART [50]. The leading cause of death among PLHIV is cardiovascular disease, which is exacerbated by hypertension [51]. Exercise is an effective non-pharmacological intervention for the prevention and treatment of hypertension [32]. CET, in particular, has been shown to lower blood pressure in people with hypertension, and it can also improve cardiovascular fitness and reduce the risk of CVD [33]. CET is an established, cost-effective intervention, and it reduces central adiposity or trunk fat leading to an improved cardiovascular profile in PLHIV [34]. In the current study, we determined whether the impact of a CET intervention could reduce the hypertension burden among PLHIV in some selected districts of rural Eastern Cape, South Africa.

In this study, all tricep, bicep, subscapular, and suprailiac skin fold measurements, as well as BMI, showed a significant difference before and after the study among the intervention group. These findings correspond with those from a similar study by Mcmurray, where there was a noticeable change in the skin folds of the participants assigned to exercise, even though they did not witness a change in BMI [52]. Furthermore, in this study, the WC decreased significantly among the intervention group compared to the control group. This decrease showed a decrease in trunk fat among participants that were allocated to CET; however, HC did not show a significant decrease. These findings were similar to those found by Armstrong and colleagues [53]. There was no significant change in ABSI and BAI among the intervention group.

Concerning hypertension, the findings in this study for both DBP and SBP showed a significant change in their values before and after the study among the intervention group. As a result, one person in the intervention group developed hypertension, while no such change occurred in the control group. This was in agreement with another study on hypertension in which brisk walking was used to reduce BP [54]. The interaction between non-smoking and not consuming alcohol led to lower BP after the study period. Similar results were found in a Chinese study that included consideration of smoking and alcohol consumption as an additional measure in the BP control [55]. We found an interaction between 1S3E and amlodipine; thus, the BP of one participant taking that combination did not decrease. HIV-infected patients must choose antihypertensive medications carefully, taking into account the potential for pharmacokinetic interactions between antiviral and antihypertensive therapies as well as the impact of these medications on certain biological parameters [56].

In this study, the logistic regression analysis showed that body mass index (BMI) and waist circumference were significantly associated with hypertension among HIV-positive individuals on ART. A 71% increase in hypertension risk was seen for every unit increase in body mass index, and a 33% increase in waist circumference risk was seen for every unit increase in BMI. These findings are consistent with previous research showing that obesity and central adiposity are strong risk factors for hypertension [57,58]. Also, Park et al. [59] found that exercise leads to a decrease in WC.

### 4.1. Strength

The current study shows that hypertension can be controlled and in some cases treated using the cheaper and effective method of CET in PLHIV.

### 4.2. Limitations

The existing study comprised certain constraints, such as the short-term duration of three months and a lower number of male participants for the test cohort. These limitations pose significant concerns towards conclusively establishing an inter-drug impact. Given that the study’s timeframe was not extensive, it is possible that protracted testing could affect overall results. Additionally, there were fewer men participating in this investigation than female counterparts. Overall, the results suggest that physical activity may help control blood pressure. More research is necessary to bolster the validity of the present findings and shed light on the mechanisms underlying the decrease in blood pressure that occurs after exercise in people with resistant hypertension.

## 5. Conclusions

To conclude, the outcomes derived from this study insinuate that BMI and waist circumference bear pivotal significance as predictors of hypertension among HIV-positive individuals receiving ART in Mthatha, South Africa. The enforcement of a comprehensive exercise training program has the potential to prove as adequately effective as a non-pharmacological solution for obstructing and managing hypertension within this particular demography while curtailing other prevalence risks linked with cardiovascular disease. This study will assist with showing clinicians a way to treat and or control hypertension among PLHIV, and provides the literature with a type of exercise that can be used to treat hypertension among PLHIV. 

Future studies should use a large sample size and an extended post-intervention monitoring period to confirm these findings empirically and assess the long-term effectiveness of fitness endeavors in treating hypertensive states in PLHIV, because perceptive evaluation necessitates sufficient statistical resources along with formidable vigilance over time.

## Figures and Tables

**Figure 1 healthcare-11-01836-f001:**
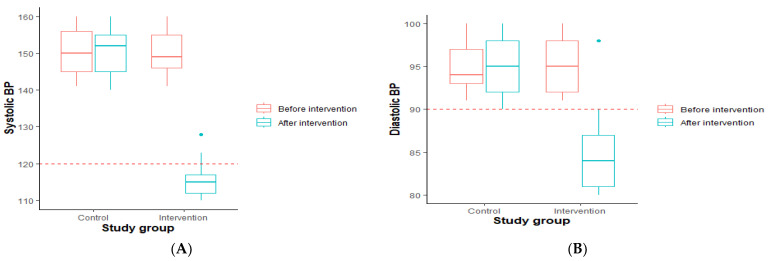
Comparison of blood pressure among the control and intervention groups before and after exercise. (**A**) Systolic blood pressure. (**B**) Diastolic blood pressure. BP: blood pressure.

**Figure 2 healthcare-11-01836-f002:**
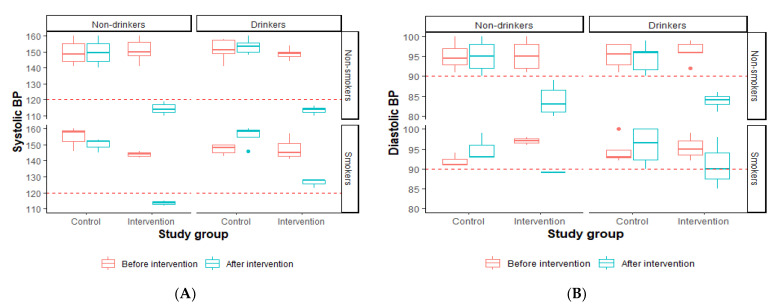
Comparison of blood pressure among the control and intervention groups before and after exercise, categorized by the participants’ smoking and drinking status. (**A**) Systolic BP. (**B**) Diastolic BP. BP: blood pressure.

**Figure 3 healthcare-11-01836-f003:**
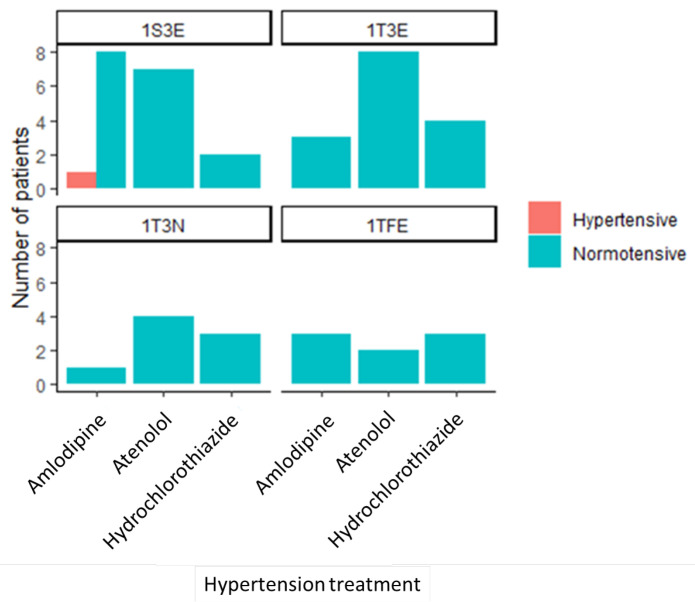
Interaction between medications used by the intervention group. Tenofovir (TDF) + Lamivudine (3TC) + Efavirenz (EFV), Stavudine (D4T) + Lamivudine (3TC) + Efavirenz (EFV), Tenofovir (TDF) + Lamivudine (3TC) + Nevirapine (NVP), and Tenofovir (TDF) + Emtricitabine (FTC) + Efavirenz (1TFE).

**Figure 4 healthcare-11-01836-f004:**
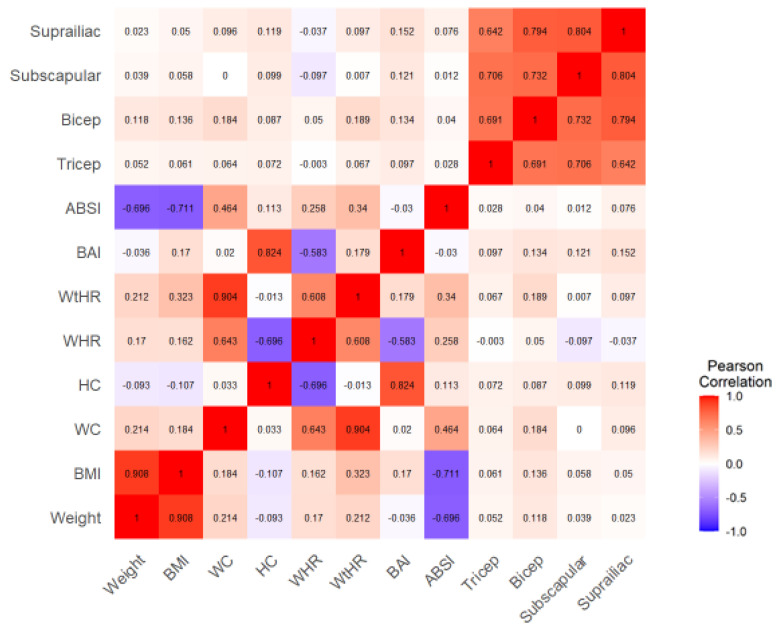
Heat map. BMI: body mass index, BAI: body adiposity index, WC: waist circumference, BP: blood pressure, ABSI: a body shape index, WHR: waist-to-hip ratio, HC: hip circumference, WtHR: waist-to-height ratio, HTN: hypertension.

**Table 1 healthcare-11-01836-t001:** Sociodemographic characteristics of study participants.

Variable	N	Overall, N = 98 ^1^	Control, N = 49 ^1^	Intervention, N = 49 ^1^	*p*-Value ^2^
Age category	98				>0.9
15–30 years		18 (18%)	9 (18%)	9 (18%)	
31–40 years		24 (24%)	12 (24%)	12 (24%)	
>40 years		56 (57%)	28 (57%)	28 (57%)	
Gender, *n* (%)	98				>0.9
Female		60 (61%)	30 (61%)	30 (61%)	
Male		38 (39%)	19 (39%)	19 (39%)	
Marital status, *n* (%)	98				0.8
Single		45 (46%)	22 (45%)	23 (47%)	
Cohabiting		2 (2.0%)	2 (4.1%)	0 (0%)	
Married		31 (32%)	14 (29%)	17 (35%)	
Divorced		8 (8.2%)	4 (8.2%)	4 (8.2%)	
Widowed		12 (12%)	7 (14%)	5 (10%)	
Education, *n* (%)	98				0.055
None		6 (6.1%)	5 (10%)	1 (2.0%)	
Primary		18 (18%)	6 (12%)	12 (24%)	
Matric		68 (69%)	37 (76%)	31 (63%)	
Tertiary		6 (6.1%)	1 (2.0%)	5 (10%)	
Employment status, *n* (%)	98				0.4
Employed		34 (35%)	19 (39%)	15 (31%)	
Unemployed		64 (65%)	30 (61%)	34 (69%)	
Smoking status, *n* (%)	98				0.5
Non-smokers		86 (88%)	42 (86%)	44 (90%)	
Smokers		12 (12%)	7 (14%)	5 (10%)	
Alcohol consumption, *n* (%)	98				0.3
Non-drinker		78 (80%)	37 (76%)	41 (84%)	
Drinker		20 (20%)	12 (24%)	8 (16%)	

^1^*n* (%); ^2^ Pearson’s Chi-squared test; Fisher’s exact test.

**Table 2 healthcare-11-01836-t002:** Physical characteristics of the study participants before and after the exercise intervention.

	Before					After				
Variable	N	Overall, N = 98 ^1^	Control, N = 49 ^1^	Intervention, N = 49 ^1^	*p*-Value ^2^	N	Overall, N = 98 ^1^	Control, N = 49 ^1^	Intervention, N = 49 ^1^	*p*-Value ^2^
Tricep (mm)	98	13 (4)	14 (4)	13 (4)	0.9	98	13 (4)	15 (4)	11 (3)	<0.001
Bicep (mm)	98	7.79 (2.45)	7.90 (2.49)	7.67 (2.42)	0.9	98	6.7 (3.1)	8.6 (1.8)	4.7 (2.9)	<0.001
Subscapular (mm)	98	19 (4)	19 (4)	19 (3)	>0.9	98	17 (3)	18 (2)	16 (3)	<0.001
Suprailiac (mm)	98	19 (4)	20 (4)	19 (3)	0.5	98	18 (3)	19 (3)	16 (4)	<0.001
Weight (kg)	98	98 (19)	97 (18)	98 (19)	0.7	98	93 (21)	101 (20)	86 (19)	0.001
BMI (kg/m^2^)	98	40 (9)	39 (8)	40 (10)	0.6	98	38 (10)	41 (9)	35 (10)	0.005
BMI category	98				>0.9	98				0.001
<25 kg/m^2^		4 (4.1%)	2 (4.1%)	2 (4.1%)			13 (13%)	1 (2.0%)	12 (24%)	
≥25 kg/m^2^		94 (96%)	47 (96%)	47 (96%)			85 (87%)	48 (98%)	37 (76%)	
WC (cm)	98	101 (9)	100 (9)	102 (9)	0.10	98	92 (12)	100 (10)	83 (9)	<0.001
HC (cm)	98	101 (12)	101 (12)	101 (12)	>0.9	98	99 (12)	101 (12)	97 (11)	0.045
WHR	98	0.98 (0.16)	1.00 (0.15)	0.96 (0.16)	0.2	98	0.94 (0.17)	1.00 (0.16)	0.88 (0.16)	<0.001
WtHR	98	0.64 (0.07)	0.63 (0.07)	0.65 (0.07)	0.076	98	0.58 (0.08)	0.63 (0.06)	0.53 (0.06)	<0.001
BAI	98	33 (7)	33 (6)	34 (7)	0.7	98	32 (7)	33 (6)	31 (7)	0.2
ABSI	98	0.72 (0.13)	0.70 (0.11)	0.74 (0.16)	0.3	98	0.67 (0.14)	0.69 (0.13)	0.65 (0.15)	0.10
SBP (mmHg)	98	150 (6)	150 (6)	150 (6)	0.6	98	133 (19)	151 (6)	115 (4)	<0.001
DBP (mmHg)	98	95 (3)	95 (3)	95 (3)	0.5	98	90 (6)	95 (3)	84 (4)	<0.001
HTN status, *n* (%)	98				>0.9	98				<0.001
Hypertensive		98(100%)	49(100%)	49(100%)			50 (51%)	49 (100%)	1 (2.0%)	
Normotensive		0(0%)	0(0%)	0(0%)			48 (49%)	0 (0%)	48 (98%)	

^1^ Mean (SD); n (%). ^2^ Wilcoxon rank sum test; Pearson’s Chi-squared test. BMI: body mass index, BAI: body adiposity index, WC: waist circumference, BP: blood pressure, ABSI: a body shape index, WHR: waist-to-hip ratio, HC: hip circumference, WtHR: waist-to-height ratio, HTN: hypertension, SBP: systolic blood pressure, DBP: diastolic blood pressure.

**Table 3 healthcare-11-01836-t003:** Logistic regression.

Variable	OR ^1^	95% CI	*p*-Value
Bicep (mm)	1.08	0.99, 1.19	0.090
Subscapular (mm)	1.42	0.95, 2.31	0.11
Suprailiac (mm)	0.86	0.60, 1.21	0.4
Body Mass Index (kg/m^2^)	1.71	1.20, 2.63	0.006
Waist Circumference (mm)	1.33	1.17, 1.61	<0.001
Body Adiposity Index	1.00	0.88, 1.12	>0.9

^1^ OR = odds ratio, CI = confidence interval.

## Data Availability

Data will be made available upon request from the corresponding author.

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
