# Peer review of "The Effect of Cardiorespiratory Exercise in the Prevention and Treatment of Hypertension among HIV-Infected Individuals on Antiretroviral Therapy in Mthatha, South Africa"

_healthcare, 2023, doi:10.3390/healthcare11131836_

Round 1
Reviewer 1 Report
The authors should give better literature review for the introduction part that support the need for this kind of study. There are many studies but authors have given reference of very few. This will improve the credibility and emphasize the need for such studies.
Figure 1, and 2 the statistical analysis is not clear. The asterisks that marks the significance appears to be just a dot and does not give the reader the impression that statistics was applied.
Figure 3 legend needs to be improved. It is not clear. The labelling in the figure can also be improved for better understanding for readers.
The focus of the paper is on the importance/role of exercise. The intervention section describing the exercises should be more elaborate. Time, weights used etc. How different age groups were assigned different exercises if it was the case. Was it monitored? By whom? Or the people performed these on their own? If so, how did authors ensure they were done regularly.
Immunological status is mentioned in methods. Blood was drawn, but the information from these blood tests is not given in the paper.
How where the physical characteristics measured? It is not described or mentioned in the methods. How is this useful or relevant?
There are limitations to this study, and authors have already explained them well.
Sample size is small and duration of the study is also quite short. While authors have acknowledged this, the data could be stronger if this could be improved.
English is fine. But few edits on sentence structure and spellings should be done.
Author Response
Thank you so much for the comments and please see attached our responses.

Reviewer 2 Report
Dear Authors,
Title Manuscript: The Effect of Cardio-Respiratory Exercise in The Prevention and Treatment of Hypertension Among HIV Infected Individuals on ART in Mthatha, South Africa
This study examined the effects of 12 weeks of cardiorespiratory exercise intervention on cardiorespiratory fitness and physiological characteristics in patients with HIV- Mthatha, South Africa. This is an important study since the study population is a group of patients with HIV but at the moment MAJOR REVISIONS are necessary in order to make it suitable for a final decision for “Healthcare”.
POINTs of STRENGTH:
1) Effects of 12 weeks of cardiorespiratory exercise intervention on cardiorespiratory fitness and physiological characteristics in patients with HIV in a RCT study;
POINTs of WEAKNESS (and/or should be revised to improve the manuscript):
Abstract:
2) The purpose of study is not specify. Please specify;
3) The, gender, age, and BMI and the type of exercise are not specify. Please specify;
4) Please report the results section based on between-group results. The significance level in the results section is unclear. Please specify;
5) Please report the “conclusion section of the Abstract” based on the results obtained from the study;
6) Please modify “keywords” as follows:
Cardiorespiratory fitness; Concurrent training; Hypertension, HIV, black people, Randomized control trial.
1. Introduction:
7) To use an abbreviation & acronym in the introduction section OR manuscript text, please write the full name in the first instance and follow it immediately by the abbreviated version in parenthesis/ brackets in the introduction section such as (HIV), and so on;
8) The hypothesis and purpose of this study can be stated in more detail;
2. Materials and Methods
2.2. Eligibility Criteria and study population definitions
9) The recruitment AND screening process of participants OR inclusion and exclusion criteria should be described in more detail such as gender, age, BMI, HIV status, hypertension OR measured resting blood pressure with empty bladder AND/OR other status, physical fitness level/VO2max, METs, drug interventions OR free of medication, and so on;
2.6. Intervention
10) Please report the exercise intervention in full detail such as the type of exercise (combined/ concurrent training OR other, exercise intensity for both aerobic and exercise interventions, exercise time, and ….);
2.7. Data Collection
11) Considering the important effects of nutrition on cardiorespiratory fitness and muscle strength parameters, did the authors monitor the nutritional status of participants during 12 weeks? IF YES, please add the method of controlling nutritional status;
12) Is this study an RCT study? IF YES, please provide all variables of RCT study for exercise interventions based on the TESTEX variables (15 items) as follows:
|
Eligibility criteria specified
|
Randomization specified
|
Allocation concealment |
Groups similar at baseline |
Blinding of all assessors |
Outcome measures assessed in 85% of patients |
Adverse events reported |
Session attendance reported |
Intent to treat analysis |
Comparison between groups-primary outcome |
Comparison between groups-secondary outcome(s) |
Point and variability measures |
Activity monitoring in control groups |
Relative exercise intensity remained constant
|
Exercise volume characteristics and energy expenditure
|
2.8.1. Hypertension
13) As mentioned above, resting blood pressure with empty bladder was measured OR other status. Please specify and report;
2.9. Data Analysis
14) The significance level of statistical analysis was considered for one-tailed? OR two-tailed? Please specify;
3. Results
15) The units of variables in Tables are unknown. Please specify;
4. Discussion and 5. Conclusions
16) As mentioned above, authors will agree that the limitation section has to be expanded. In contrast, although authors correctly reported the some limitations of this study, they did not give any strength, which I encourage;
17) Please provide clinical perspectives for this manuscript;
18) What does this study add to the literature? Please explain.
References
19) “References” section is not always in accordance with the authors' guidelines. In particular, please check No. 2, 3-5, 11, 14, 15, 28, 30, 38, 39, and 52 for validation.
In general, this manuscript suffers from typographical errors - please organize a proper proof read by a native speaking person before you submit your revision. For example, 2.2. Eligibility Criteria and study population “defintions” OR 2.8. Definitions of “varibles”, and so on;
Best Regards
16 May 2023
In general, this manuscript suffers from typographical errors - please organize a proper proof read by a native speaking person before you submit your revision. For example, 2.2. Eligibility Criteria and study population “defintions” OR 2.8. Definitions of “varibles”, and so on;
Author Response
Thank you so much for the comments provided and please see attached our responses.

Round 2
Reviewer 2 Report
Dear Authors,
Manuscript ID: healthcare-2412185
Title Manuscript: The Effect of Cardio-Respiratory Exercise in The Prevention and Treatment of Hypertension Among HIV Infected Individuals on ART in Mthatha, South Africa
I am very grateful to the authors for their efforts.
In general, although this manuscript has found good content after correcting major revisions and major revisions are modified and added in this manuscript, several concerns and/or MINOR REVISIONS have to be addressed before a final version can be made:
POINTs of WEAKNESS (and/or should be revised to improve the manuscript):
Abstract:
1) The purpose of study is not specify. Please specify;
2) The, gender, age, and BMI and the type of exercise are not specify. Please specify;
3) The significance level in the results section is unclear. Please specify;
4) Please report the “conclusion section of the Abstract” based on the results obtained from the study;
2. Materials and Methods
2.9. Data Analysis
5) To use an abbreviation & acronym in the Data Analysis section OR manuscript text, please write the full name in the first instance and follow it immediately by the abbreviated version in parenthesis/ brackets in the “Data Analysis” section such as (AOR), and so on;
3. Results
6) The units of variables in Tables are unknown. Please specify;
7) In general, this manuscript suffers from typographical errors - please organize a proper proof read by a native speaking person before you submit your revision. For example, “The findings of Mehdad et al. t”, OR “2.10..Validity and Reliability” and so on;
Best Regards
9 June 2023
In general, this manuscript suffers from typographical errors - please organize a proper proof read by a native speaking person before you submit your revision. For example, “The findings of Mehdad et al. t”, OR “2.10..Validity and Reliability” and so on;
Author Response
Dear Authors,
Manuscript ID: healthcare-2412185
Title Manuscript: The Effect of Cardio-Respiratory Exercise in The Prevention and Treatment of Hypertension Among HIV Infected Individuals on ART in Mthatha, South Africa
I am very grateful to the authors for their efforts.
In general, although this manuscript has found good content after correcting major revisions and major revisions are modified and added in this manuscript, several concerns and/or MINOR REVISIONS have to be addressed before a final version can be made:
POINTs of WEAKNESS (and/or should be revised to improve the manuscript):
Abstract:
1) The purpose of study is not specify. Please specify;
Thank for the review, we have specified the purpose of the study in lines 17 - 19
2) The, gender, age, and BMI and the type of exercise are not specify. Please specify;
We limited by word count but however have specified the gender, age in lines 21; BMI in line 26, and type of exercise in line 15.
3) The significance level in the results section is unclear. Please specify;
We have specified the significance in line 27.
4) Please report the “conclusion section of the Abstract” based on the results obtained from the study;
We have rewritten the conclusion section of the abstract based on the results obtained from the study in lines 27-30.
- Materials and Methods
2.9. Data Analysis
5) To use an abbreviation & acronym in the Data Analysis section OR manuscript text, please write the full name in the first instance and follow it immediately by the abbreviated version in parenthesis/ brackets in the “Data Analysis” section such as (AOR), and so on;
Thank for the review, we have included the full name in the first instance and followed it immediately with the abbreviated version in parenthesis/ brackets.
- Results
6) The units of variables in Tables are unknown. Please specify;
Thank you for the review, we have included some units in the tables.
7) In general, this manuscript suffers from typographical errors - please organize a proper proof read by a native speaking person before you submit your revision. For example, “The findings of Mehdad et al. t”, OR “2.10..Validity and Reliability” and so on;
Thank you for the review, we have corrected some typo errors.